# TIMBA: Time series Imputation with Bi-directional Mamba Blocks and Diffusion Models

## Abstract

The problem of imputing multivariate time series spans a wide range of fields, from clinical healthcare to multi-sensor systems. Initially, Recurrent Neural Networks (RNNs) were employed for this task; however, their error accumulation issues led to the adoption of Transformers, leveraging attention mechanisms to mitigate these problems. Concurrently, the promising results of diffusion models in capturing original distributions have positioned them at the forefront of current research, often in conjunction with Transformers. In this paper, we propose replacing time-oriented Transformers with State-Space Models (SSM), which are better suited for temporal data modeling. Specifically, we utilize the latest SSM variant, S6, which incorporates attention-like mechanisms. By embedding S6 within Mamba blocks, we develop a model that integrates SSM, Graph Neural Networks, and node-oriented Transformers to achieve enhanced spatiotemporal representations. Implementing these architectural modifications, previously unexplored in this field, we present Time series Imputation with Bi-directional mamba blocks and diffusion models (TIMBA). TIMBA achieves superior performance in almost all benchmark scenarios and performs comparably in others across a diverse range of missing value situations and three real-world datasets. We also evaluate how the performance of our model varies with different amounts of missing values and analyse its performance on downstream tasks. In addition, we provide the original code to replicate the results.

## 1 Introduction

The issue of missing values is widely recognized and can arise from various causes, including difficulties in data collection, problems with recording mechanisms, or human errors. In the field of Multivariate Time Series (MTS) analysis, where multiple variables are recorded, sometimes with irregular frequency, this problem is even more pronounced. It affects many fields, from defective sensor records that transmit erroneous data (Wu et al., 2020), to the Internet of Things (IoT) where these failures are even more common (Ahmed et al., 2022), and even clinical contexts (Moor et al., 2020). Properly filling these gaps is crucial, as failure to do so can distort the data distribution and reduce model performance (Solís-García et al., 2023).

In recent years, this problem has been tackled using Deep Learning (DL) techniques, employing Recurrent Neural Networks (RNNs) to capture temporal representations and Graph Neural Networks (GNNs) for spatial ones (Cini et al., 2022). However, the error accumulation problem in RNNs has led to their replacement by Transformers (Tashiro et al., 2021), which use attention mechanisms to avoid these issues. Furthermore, the latest state-of-the-art results have been achieved using diffusion models (Tashiro et al., 2021; Liu et al., 2023), which excel at modeling multiple distributions in the original data. Additionally, the use of State-Space Models (SSMs) has expanded in recent years, thanks to their discretization, as presented in S4 (Gu et al., 2022b). These models, reminiscent of RNNs, have shown a high capacity to handle time series and long sequences, such as those found in speech processing, text, and even DNA sequencing (Gu et al., 2022b). This architecture has been briefly explored in the context of multivariate time series imputation (MTSI) (Alcaraz & Strodthoff, 2023). Recently, S6, an improvement over S4 that incorporates an attention-like mechanism, was developed to potentially eliminate error accumulation issues, as demonstrated in selective copy tasks

and speech processing (Gu & Dao, 2023). However, the S6 and Mamba blocks have not yet been tested in the context of MTSI.

Starting from the state-of-the-art models for MTSI, CSDI and PriSTI (Tashiro et al., 2021; Liu et al., 2023), in this article we replaced time-oriented Transformers in their architecture with S6 layers embedded within bi-directional Mamba blocks (Gu & Dao, 2023), carefully maintaining a similar parameter count for fair comparability. This combination results in our proposed method, *Time series Imputation with bi-directional Mamba Blocks and Diffusion models* (TIMBA).

Our main contributions are as follows: 1) We propose TIMBA, a model that replaces the time-oriented transformers in state-of-the-art diffusion models with bi-directional Mamba blocks, an approach not previously explored to the best of our knowledge. 2) Through an extensive benchmark using three real-world datasets, we demonstrate that TIMBA consistently achieves superior performance in almost all scenarios and performs comparably in others when compared to state-of-the-art models 3) Additionally, we expand the model's analysis by presenting an ablation study, evaluating its sensitivity to different missing rates, and using it for data imputation followed by an evaluation on a downstream task. The rest of the paper is organized as follows: In Section 2, we review related work in the literature. Section 3 introduces the mathematical background supporting our approach. Section 4 presents TIMBA. In Section 5, we discuss the benchmark, experimental setup, and results obtained. Finally, Section 6 provides the conclusion. Additional information about the code and data can be found in Appendix A.

## 2 RELATED WORKS

**Multivariate time series imputation**   MTSI has been explored through various approaches. Traditional solutions such as mean, zero, or linear imputation have been utilized to address missing values (Moor et al., 2020). However, these methods often distort the original data distribution, thus degrading the data quality. In contrast, Machine Learning (ML) offers a range of simple to sophisticated techniques, including k nearest neighbors (Beretta & Santaniello, 2016), linear predictors (Seaman et al., 2012), matrix factorization (MF) (Cichocki & Phan, 2009), vector autoregressive model-imputation (VAR) (Bashir & Wei, 2018), and multivariate Gaussian processes (MGP) (Li & Marlin, 2016), which have improved outcomes in many cases.

With the advent of DL, MTSI applications were soon discovered. Initially, Recurrent Neural Networks (RNNs) (Suo et al., 2019; Lipton et al., 2016) excelled due to their robust capacity to extract accurate temporal representations, crucial for effective imputation. Advancements in RNNs led to the adoption of architectures such as Bi-directional RNNs (BiRNNs), exemplified by BRITS (Cao et al., 2018), which analyze time series data both forwards and backwards, thereby enhancing imputation capabilities. However, recent years have seen the rise of Transformers as a dominant architecture, highlighting the limitations of RNNs. The attention mechanism in Transformers is particularly effective in identifying significant or real samples in time series, addressing the error accumulation issues prevalent in RNNs, as evidenced by several studies (Tashiro et al., 2021; Liu et al., 2023).

Lastly, the role of GNNs, which are based on graph theory, must be noted for their profound ability to extract spatial relationships among variables. Many researchers argue that for accurate imputation, it is essential to derive robust spatio-temporal representations, hence the integration of RNNs or Transformers with GNNs has become a focal point (Cini et al., 2022; Liu et al., 2023). To further enhance the effectiveness of MTSI, generative models and State-Space Models have emerged as powerful tools, offering unique advantages in handling complex data distributions and dynamics.

**Generative models for imputation**   Recently, there has been growing interest in applying generative methods to MTSI due to their ability to learn the original data distribution, making them highly suitable for addressing this issue. Variational Autoencoders (VAE) have been used in this context for several years (Fortuin et al., 2020). However, interest in generative techniques for imputation exploded following the introduction of Generative Adversarial Networks (GANs), with the ground-breaking work of Generative Adversarial Imputation Networks (GAIN) (Yoon et al., 2018) marking a significant milestone. Following this development, numerous GAN applications emerged in MTSI, often incorporating RNN layers to analyze temporal sequences (Luo et al., 2018; Miao et al., 2021).

Nevertheless, the issue of model collapse in GANs (Thanh-Tung & Tran, 2020), coupled with the emergence of Denoising Diffusion Probabilistic Models (DDPMs) and their superior ability to capture diverse and high-quality distributions, has led to a shift away from GANs. Research such as CSDI by Tashiro et al. (2021) demonstrated how DDPMs, with the right conditional information, could efficiently address MTSI challenges. Further studies, such as those by Yun et al. (2023), confirmed that unconditional DDPMs might fail, whereas Liu et al. (2023) enhanced DDPMs' ability to generate better conditional information through specialized layers, furthering their effectiveness in MTSI applications.

**State-Space Models**   State-Space Models (SSMs) (Hangos et al., 2006), derived from control theory to model system dynamics, have been successfully adapted to the Deep Learning (DL) realm as Structured State Space Models (S4) (Gu et al., 2022b). These models, similar to RNNs, have recently demonstrated their proficiency in modeling diverse sequences, leading to their application in text, audio, and even video-related problems (Gu et al., 2022a). Like RNNs, however, they lack an attention mechanism, which is crucial for avoiding error accumulation during tasks such as imputation.

Nevertheless, the latest iteration, Selective State Space Models (S6), has incorporated an attention mechanism into the SSM structure, sparking a surge in applications. These models have been employed in selective copy tasks, language modeling, DNA modeling, and audio modeling and generation (Gu & Dao, 2023), occasionally outperforming traditional transformers. Specifically, in the field of time series imputation with DDPMs, the application of S4 has so far been limited to the study by (Alcaraz & Strodthoff, 2023), while S6 remains unexplored.

## 3   BACKGROUND

### 3.1   MULTIVARIATE TIME SERIES IMPUTATION

A multivariate time series (MTS) consists of $N_t$ variables or channels recorded at different time instants denoted by $t$. Consequently, these data can be represented as $\boldsymbol{X}_t \in \mathbb{R}^{N_t \times d}$, where each row $i$ contains the $d$-dimensional vector $x_t^i \in \mathbb{R}^d$, associating the $i$th variable with the time instant $t$. Additionally, a matrix $\boldsymbol{M}_t \in \{0, 1\}^{N_t \times d}$ is used, where each entry contains a 0 if the corresponding value in $\boldsymbol{X}_t$ is missing, and a 1 if it is an observed data point.

Considering this, we define $\widetilde{\boldsymbol{X}_t}$ as the unknown ground truth variable-measure matrix, i.e., the complete time series without missing data; $\widehat{\boldsymbol{X}_t}$ denotes the time series imputed by the model, and $\mathcal{X}$ the series imputed by linear interpolation technique.

Finally, given that our work will focus on graph-based models, we model the time series as a sequence of graphs following the approach established by Cini et al. (2022). In this approach, each instant $t$ is defined as a graph $\mathcal{G}_t$ with $N_t$ nodes at each time instant. Moreover, the graph is defined as $\mathcal{G}_t = \langle \boldsymbol{X}_t, \mathcal{A}_t \rangle$, where $\boldsymbol{X}_t$ represents the data matrix and $\mathcal{A}_t$ represents the adjacency matrix at instant $t$. However, this paper only considers the approach in which the graph topology does not change over time, thus $\mathcal{A}$ is constant for each $t$.

### 3.2   DENOISING DIFFUSION PROBABILISTIC MODELS

Denoising Diffusion Probabilistic Models (DDPM) rely on a Markov chain with time steps $t = 1, \ldots, T$. Beginning with an initial state $x_T \sim \mathcal{N}(0, 1)$, these models aim to reverse the sequence of latent variables $x_t$ in the same space as $x_0$, to learn $p_\theta(x_0)$ that approximates the original data distribution $q(x_0)$ (Ho et al., 2020).

The forward process or diffusion process involves progressively corrupting an initial sample $x_0$ by gradually adding Gaussian noise until reaching $x_T \sim \mathcal{N}(0, 1)$. Noise addition is moderated by a variance scheduler $\beta_t = \beta_1, \ldots, \beta_T$ to ensure distribution variance remains controlled and converges appropriately. This process is formalized by:

$$q(x_{1:T}|x_0) := \prod_{t=1}^{T} q(x_t|x_{t-1}), \quad q(x_t|x_{t-1}) := \mathcal{N}(x_t; \sqrt{1-\beta_t}x_{t-1}, \beta_t I) \tag{1}$$

Reparameterizing the parameters from Equation 1, we define $\overline{\alpha}_t = 1 - \beta_t$ and $\alpha_t := \prod_{t=1}^{T} \overline{\alpha}_t$. Utilizing the transformation $\mathcal{N}(\mu, \sigma^2) = \mu + \sigma \cdot \epsilon$, where $\epsilon \sim \mathcal{N}(0, 1)$, allows us to derive a simplified model: $q(x_t|x_0) = \sqrt{\overline{\alpha}_t}x_0 + \sqrt{1-\overline{\alpha}_t}\epsilon$.

The reverse process, to recover the initial sample $x_0$, is defined by the following Markov chain:

$$p_\theta(x_{0:T}) := p_\theta(x_T) \prod_{t=1}^{T} p_\theta(x_{t-1}|x_t), \quad x_T \sim \mathcal{N}(0, 1) \tag{2}$$

$$p_\theta(x_{t-1}|x_t) := \mathcal{N}(x_{t-1}; \mu_\theta(x_t, t), \sigma_\theta(x_t, t)I)$$

As in (Ho et al., 2020), we do not predict $\sigma_\theta$ as it remains fixed, controlled by the scheduler. $\mu_\theta$ is recalculated as:

$$\mu_\theta(x_t, t) = \frac{1}{\sqrt{\alpha_t}}(x_t - \frac{\beta_t}{\sqrt{1-\overline{\alpha}_t}}\epsilon_\theta(x_t, t)) \tag{3}$$

With the above considerations, in order to learn to model the original distribution, for each step of the chain $t$, and an $x_t$ generated according to $q(x_t|x_0)$, our model must solve the following optimization problem defined by $\mathcal{L}(\theta) := \mathbb{E}_{t,x_o,\epsilon}[||\epsilon - \epsilon_\theta(x_t, t)||^2]$.

### 3.3 STATE-SPACE MODELS AND MAMBA

State-Space Models (SSMs) are based on the theory of continuous control systems and process an input signal as defined in Equation 4 (Brogan, 1974).

$$\dot{h}(t) = \boldsymbol{A}h(t) + \boldsymbol{B}x(t)$$
$$y(t) = \boldsymbol{C}h(t) + \boldsymbol{D}x(t) \tag{4}$$

An input signal $x(t)$ is mapped to a latent state $h(t)$ and then projected to an output signal $y(t)$. The matrices $\boldsymbol{A}$, $\boldsymbol{B}$, $\boldsymbol{C}$, and $\boldsymbol{D}$ determine the transformations involved in this process. Structured State Space Models (S4) (Gu et al., 2022b) adapt SSMs by discretizing them. This involves modeling $\boldsymbol{D}$ as a skip-connection layer, defining a step size $\Delta = t_{n+1} - t_n$, and discretizing all matrices, where $\overline{\boldsymbol{C}} = \boldsymbol{C}, \overline{\boldsymbol{D}} = \boldsymbol{D}, \overline{\boldsymbol{A}} = (\boldsymbol{I} - \Delta/2 \cdot \boldsymbol{A})^{-1}(\boldsymbol{I} + \Delta/2 \cdot \boldsymbol{A})$, and $\overline{\boldsymbol{B}} = (\boldsymbol{I} - \Delta/2 \cdot \boldsymbol{A})^{-1}\Delta\boldsymbol{B}$. This results in Equation 5, noting that $\boldsymbol{D}$ is not represented as it is modeled as a skip-connection.

$$h_t = \overline{\boldsymbol{A}}h_{t-1} + \overline{\boldsymbol{B}}x_t$$
$$y_t = \overline{\boldsymbol{C}}h_t \tag{5}$$

Following S4, Selective State Space Models (S6) were introduced (Gu & Dao, 2023), improving on S4 by making $\Delta$, $\boldsymbol{B}$, and $\boldsymbol{C}$ dependent on a selection mechanism such that $\Delta \leftarrow \tau_\Delta(\text{Parameter} + s_\Delta(x))$, $\boldsymbol{B} \leftarrow s_B(x)$ and $\boldsymbol{C} \leftarrow s_C(x)$, where $s_B(x)$, $s_C(x)$ and $s_\Delta(x)$ are implemented as parameterized linear projections focusing on the hidden state of each $t$ in the original sequence $x$.

Finally, in Gu & Dao (2023), S6 is embedded within a Mamba block, which essentially consists of Multi-layer Perceptron (MLP) layers that serve to increase the hidden state size and generate two distinct information channels. In the first channel, a convolutional layer and S6 are applied, while the second channel applies a non-linear activation function to create a gate controlling the retention of information. Lastly, a MLP layer compresses everything back to its original dimensions.

## 4 TIMBA

Our approach builds upon the architecture established by Tashiro et al. (2021) and Liu et al. (2023), which is centered on a diffusion model incorporating various blocks for refining data imputation.

### 4.1 TRAINING DDPM FOR IMPUTATION

Although the general theory behind DDPMs was discussed in Section 3.2, these models need conditional information to perform effectively for general imputation, as proposed by Tashiro et al. (2021) and demonstrated by Yun et al. (2023).

To train a DDPM for MTSI, key modifications include adapting the reverse process to incorporate this conditional information. Thus, we revise Equation 2 to include our conditional variables, which are $\mathcal{X}$ and $\mathcal{A}$:

$$p_\theta(x_{0:T}) := p_\theta(x_T) \prod_{t=1}^{T} p_\theta(x_{t-1}|x_t, \mathcal{X}, \mathcal{A})$$

$$p_\theta(x_{t-1}|x_t, \mathcal{X}, \mathcal{A}) := \mathcal{N}(x_{t-1}; \mu_\theta(x_t, \mathcal{X}, \mathcal{A}, t), \beta_t I)$$

(6)

Following this, the optimization objective will be similar to that described in Section 3.2. However, as in Tashiro et al. (2021), we need to generate synthetic missing values (generate imputation targets $x_0 \in M^{ta}$) where the imputation error is calculated. The model generates imputations for all missing values, but we filter the results and retain only those specifically marked as targets for error calculation. The optimization function then changes to that defined by Equation 7. Finally, it should be noted that there are many strategies for generating $\boldsymbol{M}^{ta}$, which are detailed further in Section 5.3.

$$\mathcal{L}(\theta) := \mathbb{E}_{t,x_o,\epsilon}[\boldsymbol{M}^{ta} \cdot ||\epsilon - \epsilon_\theta(x_t, \mathcal{X}, \mathcal{A}, t)||^2]$$

(7)

### 4.2 MODEL ARCHITECTURE

Our architecture builds upon the foundational design by Tashiro et al. (2021) and incorporates enhancements introduced by Liu et al. (2023). Referring to Figure 1, and following the information flow from left to right, we begin with the Conditional Feature Extraction Module (CFEM) (Liu et al., 2023). This module is responsible for generating conditional information that aids other components in improving imputation quality. The CFEM takes $\mathcal{X}$ and $\mathcal{A}$ as inputs and processes temporal data with bidirectional Mamba blocks, replacing the original transformers—a modification we propose and will justify subsequently. Spatial data is processed using transformer and Message Passing Neural Network (MPNN) blocks, inspired by Wu et al. (2019). This information is then refined with an MLP to produce $H^{pri}$, as illustrated in the CFEM bubble of Figure 1. The operation is defined as: $H^{pri} = \text{CFEM}(\mathcal{X}, \mathcal{A})$.

Afterwards, a series of Noise Estimation Modules (NEM) (Tashiro et al., 2021) is employed to estimate the final noise. These modules receive $H^{pri}$, the diffusion time embedding $t$, $\mathcal{A}$, and $H^{in}$. For the initial NEM, $H^{in}$ is a concatenation of noisy $\boldsymbol{X}_t$ and $\mathcal{X}$. The output generates two information flows: one as input for the next NEM block (acting as $H^{in}$) and another as $H^{out}$, serving as the initial noise estimation. Outputs from each NEM are aggregated and processed by a convolutional layer to finalize the noise estimation. The internal workings of these blocks are akin to CFEM, as depicted in the NEM bubble in Figure 1, and described by: $H^{out} = \text{NEM}(H^{in}, H^{pri}, \mathcal{A}, t)$.

Now, we will discuss our decision to replace the transformer blocks, which focus on the temporal dimension, with Mamba blocks. Originally, this architecture utilized transformers for the temporal processing of information. However, despite their known advantages in MTSI, transformers lack an intrinsic inductive bias for temporal data. In contrast, Mamba blocks provide this bias while incorporating attention mechanisms, making them more suitable for accurately capturing temporal relationships. Therefore, we propose replacing time-focused transformers with Mamba blocks.

Furthermore, the mechanism within Mamba blocks' S6 layers resembles that of RNNs. This similarity allows us to adopt a BiRNN-like approach, where each Mamba block processes time series bidirectionally. Inspired by the Vision Mamba block (Zhu et al., 2024), which processes image patches in both directions, we have modified this approach to work effectively with NEM and CFEM blocks.

Our bidirectional block, depicted in Figure 2, processes a data tensor. If it is a NEM block, it concatenates $H^{pri}$ and applies layer normalization. Dropout is employed for regularization, and

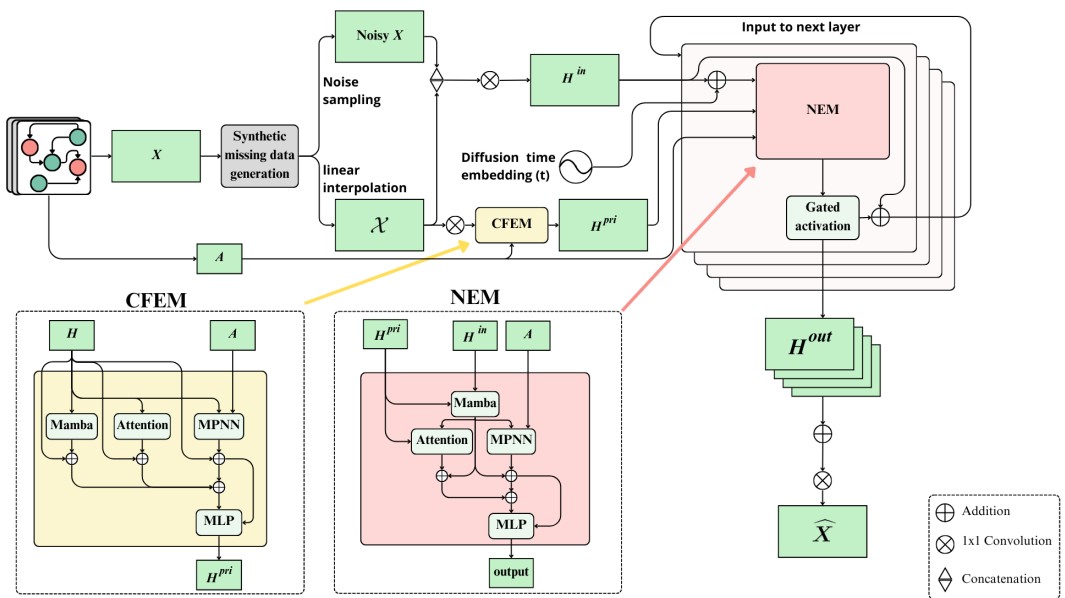

Figure 1: Architecture of the TIMBA model

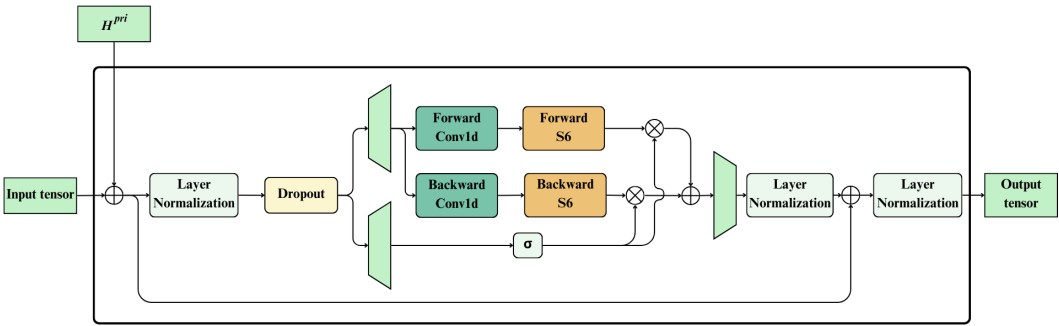

Figure 2: Implementation of the Mamba block within the NEM module capable of bi-directional time series analysis.

MLP layers expand the hidden state to create two information channels, similar to the original Mamba block. The first channel is modified to establish two additional paths that analyze the time series forwards and backwards, each with its convolutional layer and S6 layer. The second channel remains unchanged, maintaining its activation and gating mechanism. Outputs are aggregated and compressed back to their original dimensions via another MLP, followed by layer normalization. Finally, a residual connection incorporates previously processed information, producing the final output of this module.

## 5 EXPERIMENTS

### 5.1 DATASETS

We employed datasets identical to those used in Cini et al. (2022), which established benchmarks later used as comparison metrics (Tashiro et al., 2021; Liu et al., 2023). The AQI-36 dataset (Yi et al., 2016) comprises air quality data from 36 stations in Beijing, recorded hourly over a year, with 13.24% missing values. We also utilized the METR-LA and PEMS-BAY datasets (Li et al., 2018), documenting traffic in Los Angeles and the San Francisco Bay Area, respectively. METR-LA includes data from 207 sensors over four months with an 8.10% missing rate, while PEMS-BAY covers 325 sensors over six months with 0.02% missing values, both sampled every five minutes.

Consistent with the Cini et al. (2022) benchmark, we replicated the adjacency matrices using thresholded Gaussian Kernels on geographical distances, as suggested in (Li et al., 2018) and (Wu et al., 2019). The same training, validation, and testing splits were used: 70%, 10%, and 20% of the data, respectively, with identical seeds. For AQI-36, testing was conducted in February, May, August, and November, as originally designated, with 10% validation samples using the same seed.

Evaluation of these datasets includes three scenarios with synthetic missing data. In the "Block missing" scenario, 5% of the data was randomly masked, and 1-4 hour blocks were masked at each sensor with a 0.15% probability; in the "Point missing" scenario, 25% of the values were randomly masked; in the last scenario, missing values are simulated with the same distribution as the original data.

## 5.2 BASELINES

We compared our proposed method against established benchmark methods originally presented in the GRIN article (Cini et al., 2022), and subsequently extended by CSDI (Tashiro et al., 2021) and PriSTI (Liu et al., 2023). To ensure the accuracy of our comparison, we replicated the experiments of CSDI and PriSTI using the parameters reported in their original papers. The only modification made was the adjustment of small differences found in the training and validation splits of the AQI-36 dataset between CSDI, PriSTI and GRIN to maintain a fair comparison. The evaluated methods are:

1) **Statistical techniques:** MEAN (using historical mean values), DA (daily averages), and KNN (geographical proximity). 2) **Machine learning algorithms:** Lin-ITP (linear interpolation), MICE (White et al., 2011) (multiple imputation), VAR (vector autoregressive model), and KF (Kalman Filter). 3) **Matrix factorization strategies:** TRMF (Yu et al., 2016) (temporal regularized matrix factorization) and BATF (Chen et al., 2019) (Bayesian augmented tensor factorization). 4) **Autoregressive models:** BRITS (Cao et al., 2018) (bidirectional RNN), MPGRU (one-step-ahead GNN-based predictor similar to DCRNN (Li et al., 2018)), GRIN (Cini et al., 2022). 5) **Generative models:** CSDI (Tashiro et al., 2021), PriSTI (Liu et al., 2023), V-RIN (Mulyadi et al., 2021) (VAE with quantified uncertainty), GP-VAE (Fortuin et al., 2020) (VAE combined with Gaussian process), and rGAIN (Yoon et al., 2018) (GAN-based imputation with a recurrent structure).

## 5.3 EXPERIMENTAL SETTINGS

For DDPMs, 100 imputations were generated per missing value, with the median serving as the final imputation. Additionally, all experiments have been repeated 3 times using different seeds. During training, four synthetic missing data generation techniques were used:

1) **Point strategy:** A random [0, 100]% of data was masked per batch. 2) **Block strategy:** Sequences of missing values of length [L/2, L] were generated with a [0, 15]% probability, plus an additional 5% missing. 3) **Historical strategy:** Real imputation masks already present in the data were used. 4) **Hybrid strategy:** Each training sample was masked using the point strategy as a first option, or block or historical strategies as a second option with a 50% probability.

For the "Point missing" scenarios described in Section 5.1, the point strategy was used. For "Block missing" scenarios, the hybrid strategy was applied with the block strategy as the secondary option. For AQI-36, the hybrid strategy with the historical strategy as the secondary option was employed.

It is important to highlight that for the benchmark comparison in Section 5.4.1, the CSDI, PriSTI, and TIMBA models were trained for the number of epochs specified in their original papers: 200 epochs for the AQI-36 dataset and 300 epochs for the Traffic datasets. However, for the remaining experiments discussed in Section 5.4, the training epochs were limited to 50 due to time constraints during the execution of the experiments.

Finally, as in the original work by Tashiro et al. (2021), the model was trained with a learning rate of $10^{-3}$, which was reduced to $10^{-4}$ after 75% of the training epochs, and again to $10^{-5}$ after 90%. Consistent with Tashiro et al. (2021), a quadratic scheduler was used for the noise scheduling. Detailed hyperparameter settings, as well as information on training and inference times, and code reproducibility, are provided in Appendix A.

Table 1: Results obtained after testing TIMBA against the benchmark established in the literature (Cini et al., 2022). The results are shown in terms of MAE and MSE.

| | AQI-36 | | METR-LA | | | | PEMS-BAY | | | |
| | Simulated failure (24.6%) | | Block-missing (16.6%) | | Point-missing (31.1%) | | Block-missing (9.2%) | | Point-missing (25.0%) | |
| Models | MAE | MSE | MAE | MSE | MAE | MSE | MAE | MSE | MAE | MSE |
|---|---|---|---|---|---|---|---|---|---|---|
| Mean | $53.48 \pm 0.00$ | $4578.08 \pm 0.00$ | $7.48 \pm 0.00$ | $139.54 \pm 0.00$ | $7.56 \pm 0.00$ | $142.22 \pm 0.00$ | $5.46 \pm 0.00$ | $87.56 \pm 0.00$ | $5.42 \pm 0.00$ | $86.59 \pm 0.00$ |
| DA | $50.51 \pm 0.00$ | $4416.10 \pm 0.00$ | $14.53 \pm 0.00$ | $445.08 \pm 0.00$ | $14.57 \pm 0.00$ | $448.66 \pm 0.00$ | $3.30 \pm 0.00$ | $43.76 \pm 0.00$ | $3.35 \pm 0.00$ | $44.50 \pm 0.00$ |
| KNN | $30.21 \pm 0.00$ | $2892.31 \pm 0.00$ | $7.79 \pm 0.00$ | $124.61 \pm 0.00$ | $7.88 \pm 0.00$ | $129.29 \pm 0.00$ | $4.30 \pm 0.00$ | $49.90 \pm 0.00$ | $4.30 \pm 0.00$ | $49.80 \pm 0.00$ |
| Lin-ITP | $14.46 \pm 0.00$ | $673.92 \pm 0.00$ | $3.26 \pm 0.00$ | $33.76 \pm 0.00$ | $2.43 \pm 0.00$ | $14.75 \pm 0.00$ | $1.54 \pm 0.00$ | $14.14 \pm 0.00$ | $0.76 \pm 0.00$ | $1.74 \pm 0.00$ |
| KF | $54.09 \pm 0.00$ | $4942.26 \pm 0.00$ | $16.75 \pm 0.00$ | $534.69 \pm 0.00$ | $16.66 \pm 0.00$ | $529.96 \pm 0.00$ | $5.64 \pm 0.00$ | $93.19 \pm 0.00$ | $5.68 \pm 0.00$ | $93.32 \pm 0.00$ |
| MICE | $30.37 \pm 0.09$ | $2594.06 \pm 7.17$ | $4.22 \pm 0.05$ | $51.07 \pm 1.25$ | $4.42 \pm 0.07$ | $55.07 \pm 1.46$ | $2.94 \pm 0.02$ | $28.28 \pm 0.37$ | $3.09 \pm 0.02$ | $31.43 \pm 0.41$ |
| VAR | $15.64 \pm 0.08$ | $833.46 \pm 13.85$ | $3.11 \pm 0.08$ | $28.00 \pm 0.76$ | $2.69 \pm 0.00$ | $21.10 \pm 0.02$ | $2.09 \pm 0.10$ | $16.06 \pm 0.73$ | $1.30 \pm 0.00$ | $6.52 \pm 0.01$ |
| TRMF | $15.46 \pm 0.06$ | $1379.05 \pm 34.83$ | $2.96 \pm 0.00$ | $22.65 \pm 0.13$ | $2.86 \pm 0.00$ | $20.39 \pm 0.02$ | $1.95 \pm 0.01$ | $11.21 \pm 0.06$ | $1.85 \pm 0.00$ | $10.03 \pm 0.00$ |
| BATF | $15.21 \pm 0.27$ | $662.87 \pm 29.55$ | $3.56 \pm 0.01$ | $35.39 \pm 0.03$ | $3.58 \pm 0.01$ | $36.05 \pm 0.02$ | $2.05 \pm 0.00$ | $14.48 \pm 0.01$ | $2.05 \pm 0.00$ | $14.90 \pm 0.06$ |
| V-RIN | $10.00 \pm 0.10$ | $838.05 \pm 24.74$ | $6.84 \pm 0.17$ | $150.08 \pm 6.13$ | $3.96 \pm 0.08$ | $49.98 \pm 1.30$ | $2.49 \pm 0.04$ | $36.12 \pm 0.66$ | $1.21 \pm 0.03$ | $6.08 \pm 0.29$ |
| GP-VAE | $25.71 \pm 0.30$ | $2589.53 \pm 59.14$ | $6.55 \pm 0.09$ | $122.33 \pm 2.05$ | $6.57 \pm 0.10$ | $127.26 \pm 3.97$ | $2.86 \pm 0.15$ | $26.80 \pm 2.10$ | $3.41 \pm 0.23$ | $38.95 \pm 4.16$ |
| rGAIN | $15.37 \pm 0.26$ | $641.92 \pm 33.89$ | $2.90 \pm 0.01$ | $21.67 \pm 0.15$ | $2.83 \pm 0.01$ | $20.03 \pm 0.09$ | $2.18 \pm 0.01$ | $13.96 \pm 0.20$ | $1.88 \pm 0.02$ | $10.37 \pm 0.20$ |
| MPGRU | $16.79 \pm 0.52$ | $1103.04 \pm 106.83$ | $2.57 \pm 0.01$ | $25.15 \pm 0.17$ | $2.44 \pm 0.00$ | $22.17 \pm 0.03$ | $1.59 \pm 0.01$ | $14.19 \pm 0.11$ | $1.11 \pm 0.00$ | $7.59 \pm 0.02$ |
| BRITS | $14.50 \pm 0.35$ | $622.36 \pm 65.16$ | $2.34 \pm 0.01$ | $17.00 \pm 0.14$ | $2.34 \pm 0.00$ | $16.46 \pm 0.05$ | $1.70 \pm 0.01$ | $10.50 \pm 0.07$ | $1.47 \pm 0.00$ | $7.94 \pm 0.03$ |
| GRIN | $12.08 \pm 0.47$ | $523.14 \pm 57.17$ | $2.03 \pm 0.00$ | $13.26 \pm 0.05$ | $1.91 \pm 0.00$ | $10.41 \pm 0.00$ | $1.14 \pm 0.01$ | $6.60 \pm 0.10$ | $0.67 \pm 0.00$ | $1.55 \pm 0.01$ |
| CSDI | $9.74 \pm 0.16$ | $388.37 \pm 11.42$ | $1.90 \pm 0.01$ | $12.27 \pm 0.18$ | $1.77 \pm 0.05$ | $9.42 \pm 0.47$ | $\mathbf{0.84} \pm 0.00$ | $\mathbf{4.06} \pm 0.04$ | $\mathbf{0.58} \pm 0.00$ | $\mathbf{1.30} \pm 0.04$ |
| PriSTI | $9.84 \pm 0.11$ | $376.11 \pm 10.62$ | $1.78 \pm 0.00$ | $10.64 \pm 0.13$ | $1.70 \pm 0.00$ | $8.47 \pm 0.04$ | $0.87 \pm 0.01$ | $4.64 \pm 0.21$ | $0.59 \pm 0.00$ | $1.61 \pm 0.03$ |
| TIMBA | $\mathbf{9.56} \pm 0.4$ | $\mathbf{352.29} \pm 5.33$ | $\mathbf{1.76} \pm 0.02$ | $\mathbf{10.36} \pm 0.34$ | $\mathbf{1.69} \pm 0.00$ | $\mathbf{8.36} \pm 0.01$ | $\mathbf{0.84} \pm 0.01$ | $4.57 \pm 0.08$ | $\mathbf{0.58} \pm 0.00$ | $1.63 \pm 0.08$ |

Aiming for precise parameter alignment in model comparisons, when we developed TIMBA, we replaced PriSTI's dedicated temporal dimension transformer with a Mamba block with the most similar parameter count possible, while maintaining the approach described in Gu & Dao (2023) of using a factor of 2 expansion on the original tensor's hidden space. Using the METR-LA dataset as a reference, TIMBA has 876,765 parameters, while PriSTI and CSDI have 797,533 and 416,305 parameters, respectively. This results in a proportionally smaller increase in capacity, as PriSTI increases its parameters by 91.5% more compared to CSDI, whereas TIMBA only increases by 9.93% compared to PriSTI.

## 5.4 RESULTS AND DISCUSSION

To evaluate the imputation performance of TIMBA, we used the Mean Absolute Error (MAE) and Mean Square Error (MSE) metrics, following the benchmark established by Cini et al. (2022).

### 5.4.1 BENCHMARK RESULTS

Table 1 presents the evaluation results of TIMBA compared to the previously defined benchmark. Our method generally outperforms previous results, consistently achieving better or at least comparable outcomes to PriSTI.

However, a more detailed analysis reveals that our method does not perform as well in the PEMS-BAY point-missing scenario, achieving results comparable to PriSTI. Interestingly, in this specific scenario, CSDI outperforms both methods. Given that the primary difference between PriSTI and TIMBA with CSDI lies in the hyperparameters of the noise scheduler, a finer adjustment of these values might further improve our model's performance.

### 5.4.2 ABLATION ANALYSIS

This section presents the results of an ablation study comparing the performance of TIMBA using both bidirectional and unidirectional Mamba blocks. The objective of this experiment was to determine to what extent bidirectional information processing is beneficial for Mamba blocks. The results are shown in Table 2.

The analysis indicates that TIMBA achieves superior performance across all scenarios when utilizing bidirectional blocks. This finding supports the necessity of these blocks for effectively extracting improved temporal representations from the data.

Table 2: Ablation results for TIMBA. TIMBA refers to the configuration utilizing bidirectional blocks, while TIMBA-Uni represents the model using only unidirectional Mamba blocks. This experiment was conducted with training limited to 50 epochs, as described in Section 5.3

| | AQI-36 | | METR-LA | | | | PEMS-BAY | | | |
| | Simulated failure (24.6%) | | Block-missing (16.6%) | | Point-missing (31.1%) | | Block-missing (9.2%) | | Point-missing (25.0%) | |
| Models | MAE | MSE | MAE | MSE | MAE | MSE | MAE | MSE | MAE | MSE |
|---|---|---|---|---|---|---|---|---|---|---|
| TIMBA-Uni | $10.18 \pm 0.06$ | $402.62 \pm 6.57$ | $1.88 \pm 0.00$ | $12.18 \pm 0.09$ | $1.79 \pm 0.01$ | $9.59 \pm 0.05$ | $0.92 \pm 0.01$ | $5.01 \pm 0.14$ | $0.64 \pm 0.00$ | $1.99 \pm 0.06$ |
| TIMBA | $\mathbf{9.66} \pm \mathbf{0.32}$ | $\mathbf{360.66} \pm \mathbf{25.02}$ | $\mathbf{1.79} \pm 0.01$ | $\mathbf{10.73} \pm 0.24$ | $\mathbf{1.71} \pm 0.01$ | $\mathbf{8.56} \pm 0.10$ | $\mathbf{0.86} \pm 0.01$ | $\mathbf{4.80} \pm 0.19$ | $\mathbf{0.59} \pm 0.01$ | $\mathbf{1.65} \pm 0.14$ |

### 5.4.3 MISSING RATE SENSITIVITY ANALYSIS

In the following experiment, we analyze the sensitivity of our model to varying levels of missing values. We evaluate CSDI, PriSTI, and TIMBA using the METR-LA dataset under the Point-missing scenario with different rates of missing data. For each model, we used the best-performing weights obtained during a previous training of 50 epochs, consistent with the description in Section 5.3. The results for the three models are presented in terms of MAE and MSE in Tables 3 and 4, respectively.

Table 3: Sensitivity analysis for different levels of missing values in the METR-LA dataset under the Point missing scenario, presented in terms of MAE.

| | MAE - METR-LA (P) | | | | | | | | |
| Models | 10% | 20% | 30% | 40% | 50% | 60% | 70% | 80% | 90% |
|---|---|---|---|---|---|---|---|---|---|
| CSDI | 1.77 | 1.82 | 1.88 | 1.96 | 2.07 | 2.20 | 2.39 | 2.70 | 3.29 |
| PriSTI | **1.64** | 1.67 | 1.71 | 1.76 | 1.81 | 1.89 | 1.99 | 2.14 | 2.43 |
| TIMBA | **1.64** | **1.66** | **1.70** | **1.75** | **1.80** | **1.88** | **1.98** | **2.13** | **2.41** |

Table 4: Sensitivity analysis for different levels of missing values in the METR-LA dataset under the Point missing scenario, presented in terms of MSE.

| | MSE - METR-LA (P) | | | | | | | | |
| Models | 10% | 20% | 30% | 40% | 50% | 60% | 70% | 80% | 90% |
|---|---|---|---|---|---|---|---|---|---|
| CSDI | 8.50 | 9.10 | 9.92 | 10.88 | 12.14 | 13.86 | 16.54 | 21.75 | 32.54 |
| PriSTI | 7.71 | 8.07 | 8.61 | 9.29 | 10.08 | 11.24 | 12.88 | 15.88 | 22.08 |
| TIMBA | **7.60** | **7.91** | **8.48** | **9.15** | **9.96** | **11.09** | **12.65** | **15.55** | **21.57** |

Analyzing the results, we find that TIMBA consistently delivers the best performance, particularly in terms of MSE. This demonstrates that our model effectively manages higher rates of missing values in the input data, further highlighting the advantages of SSMs over classical Transformers for these time series tasks.

### 5.4.4 DOWNSTREAM TASK ANALYSIS

Evaluating imputation models requires assessing their effectiveness on downstream tasks (Wang et al., 2024). This section outlines a specific task using imputed data to evaluate CSDI, PriSTI, and TIMBA. The downstream task designed for this experiment involves predicting the value of a node at time $t$ using the values of other nodes at the same time $t$.

For this section, we performed the node forecasting task using two different nodes. We conducted all experiments twice, once targeting node 14 and once targeting node 31, to ensure a more informative comparison. These nodes correspond to stations with the highest and lowest connectivity in the graph, following a similar approach to Cini et al. (2022) in selecting nodes for experiments.

We use the AQI-36 dataset, imputing missing values with the best weights obtained from the results in Table 1. For this task, we only impute validation and test data to ensure that the imputation models

work with data they have never seen. These new imputed data are split into a new training set (80%) and test set (20%), normalized using a MinMax scaler.

We employ an MLP with one hidden layer of 100 neurons and an output layer of 1. This network is trained for 500 epochs to minimize the MSE between the actual and predicted values, using the same five seeds for the imputed data with CSDI, PriSTI, and TIMBA. Final test results are measured using MSE and MAE and only consider actual values. Imputed target values are excluded from the final test results for the calculation of both metrics.

Table 5: Results of the downstream task for node value prediction using data imputed by CSDI, PriSTI, and TIMBA.

| | Sensor 14 | | Sensor 31 | |
|---|---|---|---|---|
| Models | MAE | MSE | MAE | MSE |
| CSDI | $6.51 \pm 0.69$ | $96.99 \pm 21.25$ | $11.99 \pm 1.92$ | $376.40 \pm 148.06$ |
| PriSTI | $6.46 \pm 0.71$ | $92.70 \pm 20.27$ | $11.70 \pm 1.80$ | $361.19 \pm 132.99$ |
| TIMBA | $\mathbf{6.45} \pm 0.69$ | $\mathbf{91.90} \pm 20.33$ | $\mathbf{11.68} \pm 1.77$ | $\mathbf{359.80} \pm 131.74$ |

As shown in Table 5, TIMBA achieves the best results for both nodes and both metrics. This indicates that the quality of imputation provided by our method is advantageous for use as a preprocessing step to improve subsequent tasks.

### 5.4.5 LIMITATIONS

Analyzing the limitations of our model, it should be noted that our method does not assume any distribution in the missing values beyond their occurrence through a stationary process. Throughout the paper, we mainly focus on the missing at random (MAR) scenario (Rubin, 1976). Beyond this, we make no assumptions about the amount or length of the missing data.

## 6 CONCLUSIONS

In this paper, we presented TIMBA, a model that replaces the time-oriented transformer layers in state-of-the-art diffusion models for multivariate time series imputation with bidirectional Mamba blocks. Our extensive benchmark with three real-world datasets demonstrates that TIMBA either surpasses or performs comparably to the current state-of-the-art. Additionally, we showed that TIMBA can scale effectively with longer temporal sequences, generally achieving better results as the number of time steps per sample increases.

For future work, it would be valuable to explore ways to further reduce training and inference time, potentially through the application of latent diffusion models or denoising diffusion implicit models. Additionally, it would be interesting to apply our architecture to time series forecasting problems.

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

# A IMPLEMENTATION DETAILS

## A.1 COMPLETE HYPERPARAMS DESCRIPTION

Table 6 lists all the hyperparameters used to obtain the results presented in this paper. Most of these hyperparameters are directly taken from the PriSTI GitHub repository to ensure a fair comparison.

Table 6: The hyperparameters of TIMBA for all datasets.

| Description | AQI-36 | METR-LA | PEMS-BAY |
|---|---|---|---|
| Batch size | 16 | 4 | 4 |
| Time length $L$ | 36 | 24 | 24 |
| Epochs | 200 | 300 | 300 |
| Learning rate | 0.001 | 0.001 | 0.001 |
| Layers of noise estimation | 4 | 4 | 4 |
| Channel size $d$ | 64 | 64 | 64 |
| Number of attention heads | 8 | 8 | 8 |
| Minimum noise level $\beta_1$ | 0.0001 | 0.0001 | 0.0001 |
| Maximum noise level $\beta_T$ | 0.2 | 0.2 | 0.2 |
| Diffusion steps $T$ | 100 | 50 | 50 |
| Number of virtual nodes $k$ | 16 | 64 | 64 |
| Mamba block dropout | 0.1 | 0.1 | 0.1 |
| SSM state expansion factor | 16 | 16 | 16 |
| Mamba local convolution width | 4 | 4 | 4 |
| Mamba block expansion factor | 2 | 2 | 2 |

To expand on this information, it is important to note that the virtual nodes $k$ were introduced by Liu et al. (2023). These nodes help reduce the complexity of transformers in handling spatial information by compressing real nodes into a defined number of virtual nodes $k$.

## A.2 CODE REPRODUCIBILITY

This project has been conducted with a focus on facilitating code reproducibility and easy access to the datasets used. The code is available on GitHub: **[Hidden for anonymization – check supplementary material during review]**.

The implementation is in Python (Van Rossum & Drake, 2009), utilizing the following open-source libraries:

- Pytorch (Paszke et al., 2017).
- Pytorch Lightning (Falcon & The PyTorch Lightning team, 2019)
- Numpy (Harris et al., 2020).
- Torch spatio-temporal (Cini & Marisca, 2022).
- Pandas (pandas development team, 2020; Wes McKinney, 2010).
- Hydra (Yadan, 2019).

Additionally, to simplify execution, a Docker image and container (Merkel, 2014) have been developed, along with scripts to create and run them easily.

All experiments were conducted on a computer with the following specifications: Ubuntu 22.04.2 LTS, AMD Ryzen Threadripper PRO 3955WX 16-Cores CPU, NVIDIA RTX A5000 24 GB GPU, and 8X16 GB (128GB) DDR4 RAM. The runtime results obtained using this setup are presented in Table 7.

Table 7: Training and inference times measured in hours.

| Models | AQI-36 Training | AQI-36 Inference | METR-LA Training | METR-LA Inference | PEMS-BAY Training | PEMS-BAY Inference |
|---|---|---|---|---|---|---|
| CSDI | 0.29 | 0.22 | 7.28 | 1.74 | 19.11 | 4.62 |
| PriSTI | 0.40 | 0.33 | 8.21 | 2.44 | 19.72 | 5.99 |
| TIMBA | 0.56 | 0.44 | 12.83 | 3.65 | 32.74 | 8.71 |

## A.3 DATA AVAILABILITY

The datasets used in this paper are publicly and freely accessible. Specifically, the Torch Spatio-Temporal library (Cini & Marisca, 2022) includes code for downloading and preparing these datasets, so there is no need to obtain them separately before running our code.

