# OpenReview forum: "TIMBA: Time series Imputation with Bi-directional Mamba Blocks and Diffusion models"
_ICLR.cc/2025/Conference — ICLR 2025 Conference Withdrawn Submission_

### Official Review · Reviewer_G76U · 2024-10-16

**Soundness:** 3
**Presentation:** 3
**Contribution:** 1
**Rating:** 3
**Confidence:** 4

**Summary:**

The paper introduces TIMBA, an architecture for time series imputation. The main idea behind the proposed approach is to replace the Transformer layers of an existing architecture (PriSTI) with bidirectional Mamba blocks. Empirical performance matches or even surpasses the state of the art.

**Strengths:**

* **Relevant problem**: Integrating SSMs into time series models is an important direction for future work.
* **Somewhat strong empirical results**: Good performance on relevant benchmarks, though improvements are limited, especially considering the reported standard deviations. Furthermore, the ablation analysis is limited (see weaknesses).

**Weaknesses:**

**Major weaknesses**

* **No clear motivation**: The authors argue that the proposed Mamba blocks should be used instead of Transformers as they are more tailored to processing time series given their inductive biases. However, the same could be said for models like RNNs and TCNs and for specialized Transformer architectures for time series. While SSMs have proven particularly useful for processing long sequences, the sequences processed here are a maximum of 36 time steps. It's unclear what specific benefits the proposed architectural modification offers. I suspect that the performance gap (often not particularly significant given the reported deviations) relative to existing models could be eliminated with better hyperparameter selection.

* **Limited novelty**: The paper suggests replacing the sequence modeling part of an existing architecture with another (existing) sequence modeling layer. The contributions, therefore, appear limited by ICLR standards.

* **Limited analysis**: Since the contribution is primarily an architectural modification, this should be the focus of additional ablation studies. Simply showing that the bidirectional version of the model achieves better results is not sufficient. One possibility would be to replace the S6 layers with RNNs/TCNs and/or replace the Mamba block entirely with other sequence modeling blocks.

**Minor comments**
* Some relevant missing references, e.g.,
    - [1] SPIN, an attention-based graph neural network for imputation, which mitigates the error accumulation seen in GRIN.
    - [2] SAITS, a popular attention-based imputation architecture.

#### References

[1] Marisca et al., "Learning to Reconstruct Missing Data from Spatiotemporal Graphs with Sparse Observations", NeurIPS 2022\
[2] Du et al., “SAITS: Self-Attention-based Imputation for Time Series”, Expert Systems with Applications, 2022

**Questions:**

Please comment on the highlighted weaknesses.

---

### Official Review · Reviewer_6mih · 2024-10-26

**Soundness:** 2
**Presentation:** 2
**Contribution:** 1
**Rating:** 3
**Confidence:** 3

**Summary:**

In this paper, the authors propose the TIMBA model for multivariate time series imputation, integrating the Diffusion Probabilistic Model (DDPM) with State-Space Models (SSM), Graph Neural Networks (GNN), and other advanced techniques. By replacing time-oriented Transformers with SSMs and embedding the Selective State Space Models (S6) architecture within Mamba blocks, TIMBA effectively combines SSMs, GNNs, and node-based Transformers to enhance spatiotemporal representations.

**Strengths:**

In this paper, the authors propose TIMBA, a novel model that replaces time-oriented transformers in diffusion models with bi-directional Mamba blocks. Through extensive benchmarking on three real-world datasets, TIMBA consistently outperforms or matches state-of-the-art models. The paper further assesses the model's robustness through an ablation study, sensitivity analysis to varying missing rates, and downstream task evaluation after data imputation.

**Weaknesses:**

1  Unclear Motivation: The paper combines several machine learning methods (e.g., Transformers, SSMs, diffusion models, GNNs), but the motivation behind each choice is unclear. There is little to no theoretical analysis to support the use of these models or to explain why this specific combination of techniques is beneficial.

2  Lack of Comparison to Relevant Baselines: The experiments do not compare the proposed model to other state-of-the-art diffusion-based models like SSSD or diffusion with RNNs. This omission leaves uncertainty about whether the proposed TIMBA models truly offer better performance than existing methods.

3 Results Discrepancies: Some of the results reported in the paper differ from those in previous works (e.g., PriSTI). However, the authors do not provide an explanation for these discrepancies. This raises concerns about whether the differences arise from the training strategy, model implementation, or some other factor.

4 Inadequate Coverage of Time Series Imputation: While the paper claims to focus on time series imputation, the experiments primarily involve spatio-temporal datasets. There are no experiments on non-spatio-temporal datasets like clinical healthcare, which limits the generalizability of the model. This mismatch between the title and the scope of experiments weakens the overall contribution.

5  Format Issues: Some tables in the paper are out of range, indicating formatting problems that need to be addressed to improve readability and presentation quality.

**Questions:**

1  Why Choose SSM over Time-Oriented Transformers?: Why does the model replace time-oriented transformers with State-Space Models (SSMs)? What makes SSMs better suited for time modeling compared to transformers?

2  Overfitting Concerns: In the results from the PEMS-Bay dataset, the CSDI model using only the diffusion mechanism slightly outperforms the proposed model, which includes additional techniques such as GNN. Does this suggest that the proposed model might be overfitting to the spatio-temporal datasets?

3  Ablation Study: The ablation study focuses only on TIMBA-Uni and TIMBA. What about the contributions of other components of the model? How do they impact performance individually?

4  Latest Works in Time Series Imputation: In 2024, many papers on time series imputation have been published. I recommend incorporating more recent works in the introduction to provide a more comprehensive and up-to-date background.
ImputeFormer: Low rankness-induced transformers for generalizable spatiotemporal imputation; SADI: Similarity-Aware Diffusion Model-Based Imputation for Incomplete Temporal EHR Data; Higher-order Spatio-temporal Physics-incorporated Graph Neural Network for Multivariate Time Series Imputation; and other recent works.

5  Training Strategy and Result Variations: Could the differences in results from prior works (e.g., PriSTI) be due to different training strategies, or are there other factors that could explain these variations?

---

### Official Review · Reviewer_Qg5b · 2024-11-02

**Soundness:** 2
**Presentation:** 2
**Contribution:** 1
**Rating:** 3
**Confidence:** 4

**Summary:**

The paper presents the TIMBA model for multivariate time series imputation (MTSI). This work builds on the idea of using diffusion models to learn the latent distribution of time series data, enabling the generation of complete sequences from incomplete data. It also replaces the time-oriented Transformer model with a bidirectional Mamba model containing S6 blocks. A GNN-based model is used to represent the time series data, which serves as conditional variables for the diffusion models and the inputs for the Conditional Feature Extraction Module (CFEM). The Noise Estimation Module (NEM) is used to model the final noise. The paper claims that this approach enhances the spatiotemporal representation for the imputation task.

Overall, the paper highlights various techniques widely used in the time-series and machine learning communities. Although it attempts to combine these techniques to solve the imputation task, it is difficult to understand the utility of each component and the relationships between them. I believe a few components could be combined to perform imputation more effectively. Therefore, this study proposes an over-designed architecture that may lack sufficient rationale to justify each component.

**Strengths:**

1. It explores some interesting technical combinations of existing models, such as bidirectional Mamba and S6, which could be beneficial for the imputation task as it requires bidirectional context.

2. The GNN model is used to capture spatial relationships between variables, which could complement temporal dependencies.

3. It examines the sensitivity to various levels of missing values in the imputation tasks.

**Weaknesses:**

1. This paper introduces a variety of existing techniques, combining them into a new model, TIMBA. However, it is primarily a combination of existing techniques and offer little in terms of new insights or techniques.

2. The technical contribution is limited. For instance, the second and the third contribution are benchmark experiments and ablation studies, which are generally not considered as contribution.

3. Although this paper focuses on diffusion and bidirectional Mamba, many components appear unnecessary. In diffusion-based time series models, a typical setup includes only a backbone model and a diffusion model. For example, the foundational work TimeGrad uses RNN as the backbone model [1]. In the imputation task, the foundational work, CSDI, employs a single-layer transformer encoder as the backbone and demonstrates that bidirectional models generally outperform unidirectional ones. Therefore, this study would likely only need bidirectional Mamba as the backbone for imputation.

    [1] Autoregressive denoising diffusion models for multivariate probabilistic time series forecasting.

    [2] CSDI: Conditional Score-based Diffusion Models for Probabilistic Time Series Imputation

4. Althought this paper proposes ablation studies as a contribution, however, the benefits of various blocks are not known. For example, the study believes that the bidirectional Mamba with S6 is a better design, while we do not know whether the S6 can improve performance. A complete ablation studies should be conducted for all involved components, helping justify the necessity of each component. For example, for your proposed Mamba architecture, it should include the ablation on Mamba with and without S6, unidirectional vs bidirectional.

5. Many blocks are not discusses and do not have clear rationale. For example, the Conditional Feature Extraction Module (CFEM) and Noise Estimation Modules (NEM) are not discussed sufficiently. It does not show the motivation behind these blocks. We also do not know why these models should be combined with other blocks.  The authors may address the following questions, but are not limited to these: what is the specific purpose of the CFEM in this architecture? How does the NEM contribute to the overall performance of the model?


6. I find the use of GNN-based representation for time series confusing, as its necessity is unclear.Could you elaborate on why GNN-based representation is necessary for this task, and what specific advantages it provides over alternative approaches?


7. Although it is positioned as a diffusion model for imputation tasks, it only compares with two baselines in this category, CSDI and PriSTI. Most of the baselines are from traditional ML methods, which may not be suitable for comparison.

**Questions:**

Same as above.

---

### Official Review · Reviewer_kwHm · 2024-11-04

**Soundness:** 3
**Presentation:** 3
**Contribution:** 2
**Rating:** 5
**Confidence:** 4

**Summary:**

This paper introduces a model called Time series Imputation with bi-directional Mamba Blocks and Diffusion models (TIMBA) to address the time series imputation. The model combines a bi-directional Mamba block and a diffusion-based generative model to do imputation for time-series data. Empirical evaluation using three benchmark datasets show improved performance against various baseline methods for time series imputation.

**Strengths:**

- Overall, the clarity of the paper is good. It is well-written and easy to follow.
- By and large, the proposed method is technically sound.
- Empirical evaluation shows improved performance against existing methods.

**Weaknesses:**

- The major weakness of the paper is the limited novelty. Using diffusion model for time series imputation has been explored in a few existing work. Particularly, MSTI (Alcaraz & Strodthoff, 2023) explored using Mamba as the backbone of diffusion for time-series data. Given the existing work, it seems that the main technical contribution is to replace S4 with another SSM variant, S6. The technical contribution looks incremental to me.
- Most experiments limit the training epochs to 50, which might not be sufficient for models to converge, especially for generative models.
- In the limitations section, it is argued that the paper addresses the missing-at-random (MAR) problem. Please clarify if the paper addresses MAR or missing-completely-at-random (MCAR). How is the probability of missingness in MAR being considered?

**Questions:**

Please see my comments in the Weaknesses section.

---

### Note · Authors · 2024-11-18

**Comment:**

We sincerely appreciate the reviews from all the reviewers and recognize that important points have been raised that we would like to address to further improve our paper. For this reason, we have decided to withdraw our submission with the goal of making these improvements. Thank you once again for your valuable feedback.

**Withdrawal Confirmation:**

I have read and agree with the venue's withdrawal policy on behalf of myself and my co-authors.